



# ClimateNet: an expert-labelled open dataset and Deep Learning architecture for enabling high-precision analyses of extreme weather

Prabhat[1,2,*], Karthik Kashinath[1,*], Mayur Mudigonda[1,10,*], Sol Kim[2], Lukas Kapp-Schwoerer[3],
Andre Graubner[3], Ege Karaismailoglu[3], Leo von Kleist[3], Thorsten Kurth[4], Annette Greiner[1],
Kevin Yang[2], Colby Lewis[2], Jiayi Chen[2], Andrew Lou[2], Sathyavat Chandran[5], Ben Toms[6],
Will Chapman[7], Katherine Dagon[8], Christine A. Shields[8], Travis O'Brien[9,1], Michael Wehner[1], and
William Collins[1,2]

*__Equal contributions__
[1]Lawrence Berkeley National Laboratory, Berkeley, CA, USA
[2]University of California, Berkeley, CA, USA
[3]ETH Zurich, Switzerland
[4]NVIDIA, Santa Clara, CA, USA
[5]Rice University, Houston, TX, USA
[6]Colorado State University, Fort Collins, CO, USA
[7]Scripps Institution of Oceanography, University of California, San Diego, La Jolla, CA, USA
[8]National Center for Atmospheric Research, Boulder, CO, USA
[9]Indiana University, Bloomington, IN, USA
[10]Terrafuse, Berkeley, CA,USA

*Correspondence to:* Karthik Kashinath (kkashinath@lbl.gov)





**Abstract.**

Identifying, detecting and localizing extreme weather events is a crucial first step in understanding how they may vary under different climate change scenarios. Pattern recognition tasks such as classification, object detection and segmentation (*i.e.* pixel-level classification) have remained challenging problems in the weather and climate sciences. While there exist many empirical heuristics for detecting extreme events, the disparities between the output of these different methods even for a single event are large and often difficult to reconcile. Given the success of Deep Learning (DL) in tackling similar problems in computer vision, we advocate a DL-based approach. DL, however, works best in the context of supervised learning; when labeled datasets are readily available. Reliable, labeled training data for extreme weather and climate events is scarce.

We create 'ClimateNet' – an open, community-sourced human expert-labeled curated dataset – that captures tropical cyclones (TCs) and atmospheric rivers (ARs) in high-resolution climate model output from a simulation of a recent historical period. We use the curated ClimateNet dataset to train a state-of-the-art DL model for pixel-level identification, *i.e.* segmentation, of TCs and ARs. We then apply the trained DL model to historical and climate change scenarios simulated by the Community Atmospheric Model (CAM5.1) and show that the DL model accurately segments the data into TCs, ARs or 'the background' at a pixel level. Further, we show how the segmentation results can be used to conduct spatially and temporally precise analytics by quantifying distributions of extreme precipitation conditioned on event types (TC or AR) at regional scales. The key contribution of this work is that it paves the way for DL-based automated, hi-fidelity and highly precise analytics of climate data using a curated expert-labelled dataset – ClimateNet.

ClimateNet and the DL-based segmentation method provide several unique capabilities: (i) they can be used to calculate a variety of TC and AR statistics at a fine-grained level; (ii) they can be applied to different climate scenarios and different datasets without tuning as they do not rely on threshold conditions; and (iii) the proposed DL method is suitable for rapidly analyzing large amounts of climate model output. While our study has been conducted for two important extreme weather patterns (TCs and ARs) in simulation datasets, we believe that this methodology can be applied to a much broader class of patterns, and applied to observational and reanalysis data products via transfer learning.





# 1 Introduction

Climate change is arguably one of the most pressing challenges facing humanity in the 21st century. Identifying weather patterns that frequently lead to extreme weather events is a crucial first step in understanding how they may vary under different climate change scenarios. To do so, climate scientists have largely relied on custom *heuristics* for the identification of these events (Hodges, 1995; Neu et al., 2013; Prabhat et al., 2015b; Shields et al., 2018a; Ullrich and Zarzycki, 2017). However, there are often large discrepancies between different detection algorithms for the same type of pattern or event. Different heuristics rely on different subsets of variables, and choices of threshold conditions. Often there are large discrepancies on the overall numbers of such events, their frequencies of occurence, intensities and spatial extents.

As an illustration of this limitation, many different atmospheric river (AR) detection algorithms exist that produce largely different outputs. This recently motivated researchers to launch the Atmospheric River Tracking Methods Intercomparison Project (ARTMIP), which found that AR counts can differ by an order of magnitude depending on which algorithm is used (Shields et al., 2018b). Similarly, the IMILAST project (Neu et al., 2013) sought to compare detection algorithms for Extra-Tropical Cyclones (ETC) and concluded that various ETC detection methods produce widely varying estimates (3-7x) for ETC counts. A related, but understudied issue pertains to heuristics for defining the *spatial extent* of weather patterns (Chavas et al., 2015; Allen and Ingram, 2002; Gao et al., 2015; Knutson et al., 2019; Patricola and Wehner, 2018) . Given the wide discrepancy in storm counts, we have limited reason to believe that the heuristics pertaining to storm extents fare any better. It is noteworthy that these issues have plagued the climate analytics and climate informatics communities for over 30 years, and it is unclear as to what the solution might be – development of yet more heuristics?, weighted combinations of heuristic output?, Bayesian or probabilistic treatment of heuristic output?, etc.

To overcome these long-standing challenges and discrepancies in the field, we turn to techniques from a different domain: namely Deep Learning (DL) from the field of computer science. The application areas of computer vision, speech recognition and robotics have struggled with custom heuristics since the mid-1980s, and have recently conclusively demonstrated that Deep Learning techniques can successfully and significantly advance the state of the art of pattern recognition and pattern discovery; both of which are critical needs of the weather and climate science communities (LeCun et al., 2015; Levine et al., 2016). Inspired by these results, recent work has demonstrated that DL can indeed be applied to identifying the type (classification), spatial extent (localization), and pixel-level masks (segmentation) of weather and climate patterns (Liu et al., 2016; Hong et al., 2017; Racah et al., 2017; Kurth et al., 2018). The success of these applications, however, was limited by the lack of high quality, reliable, labeled data which is a key requirement for the success of supervised DL techniques. The fields of weather and climate science currently lack these crucial expert-labeled datasets.

Scientists in both of these fields have been increasingly adopting the use of ML and DL, owing in part to the increase in available computational power and the ever growing volumes of data due to rapid and significant increases in temporal and spatial resolution of climate models, reanalysis products and observational datasets. ML and DL techniques, many of which were developed to work with 'big data', have recently shown great promise in applications in meteorology and climate: parameterization in climate models, post-model bias correction, and forecasting of the El Niño Southern Oscillation (ENSO) and



Madden-Julian Oscillation (MJO) (O'Gorman and Dwyer (2018); Brenowitz and Bretherton (2018); Chapman et al. (2019); Mahesh et al.; Ham et al. (2019); Toms et al. (2019); McGovern et al. (2017)). To highlight one success, Ham et al. (2019) demonstrated the skill of DL on forecasting El Niño states and found DL to forecast with superior leadtimes than state-of-the-art dynamical models. Many of the ML and DL techniques used, again rely on the availability and quality of labeled data. Some
of the aforementioned papers utilize specific ENSO or MJO indices which have rigid and established definitions (e.g. Niño3.4) which allows for straightforward generation of labels (ENSO states) which can be used for training the DL model. However, even these large-scale modes have variety in their definition (NOAA (2019)). A major limitation to expanding the success of DL to a greater variety of weather and/or climate phenomena is the lack of large reliable high-quality labeled datasets.

Given: (i) the ambiguities of existing heuristics of detecting weather and climate patterns; (ii) the power of DL in recogniz-
ing complex patterns *without* requiring engineered features; (iii) the scarcity of reliable labeled data; and (iv) the increasing relevance of ML and DL to weather and climate science; we have developed 'ClimateNet' – a community-sourced, human expert-labeling strategy to prepare a vast and reliable database of weather and climate pattern labels to push the frontier of DL methods for a variety of important and urgent pattern recognition tasks in the weather and climate sciences. Here we construct datasets which capture the boundaries of two important intense storm patterns, Tropical Cyclones (TCs) and Atmospheric
Rivers (ARs) and we envision expanding ClimateNet to include many other weather and climate events.



## 2 ClimateNet Dataset

The first step towards building an expert-labeled dataset is the development of a labeling interface, whereby climate data can be ingested and climate experts can annotate events of interest, such as atmospheric rivers and tropical cyclones. The requirements for such an interface are: (i) sufficient information to annotate events correctly; (ii) ability to add, delete and
modify labels easily, and (iii) facility to specify user confidence for each label individually.

### 2.1 ClimateContours

We develop the ClimateContours tool, which is a guided user interface for annotating climate events. ClimateContours is built upon the annotation tool LabelMe (Russell et al., 2008), which was originally developed to aid the generation of annotated examples for training supervised learning models in the computer vision community. ClimateContours is a versatile and
easy-to-use tool, hosted at *http://labelmegold.services.nersc.gov/climatecontours_gold/tool.html*, which leverages the science gateway infrastructure at the National Energy Research Supercomputer Center (NERSC). ClimateContours renders snapshots from a prescribed climate dataset and allows the user to label two types of events - Atmospheric Rivers (ARs) and Tropical Cyclones (TCs). The labeler chooses the pen-like tool to manually place vertices of a polygon around an event of choice. The placement of vertices ceases when a convex hull is created, *i.e.* when the last vertex coincides with the first vertex. The labeler
then chooses the type of event (AR, TC) and the confidence of their labeling process (high, medium, and low).

The labeler has the option to delete edges or the entire polygon and re-create polygons as many number of times as they wish. In addition, a labeler may zoom in to view events at a finer scale and switch between various views of raw and derived variables to help inform their labeling.

Currently, ClimateContours renders snapshots from 25-km CAM5.1 climate model output (Wehner et al., 2014). We choose
this particular model for it's high resolution, high fidelity for simulating tropical cyclones and atmospheric rivers, and for the large amount of readily available output data for multiple climate change scenarios, thus making training DL models and testing their generalization capabilities viable. Output from this model contains dozens of physical variables, such as wind velocity, temperature, pressure, and humidity at different vertical levels and across the globe (3 spatial dimensions and time). These variables contain information relevant to the dynamics of weather and climate phenomena, but not all variables are needed to
detect a weather event. Based on the experience and wealth of knowledge accumulated by meteorologists, and weather and climate scientists, and for relative ease of use, we provide a subset of six variables - in various combinations - to the user to aid them in creating labels for TCs and ARs through ClimateContours. These are the leading variables that are used to define and characterize TCs and ARs and are shown in the following table:





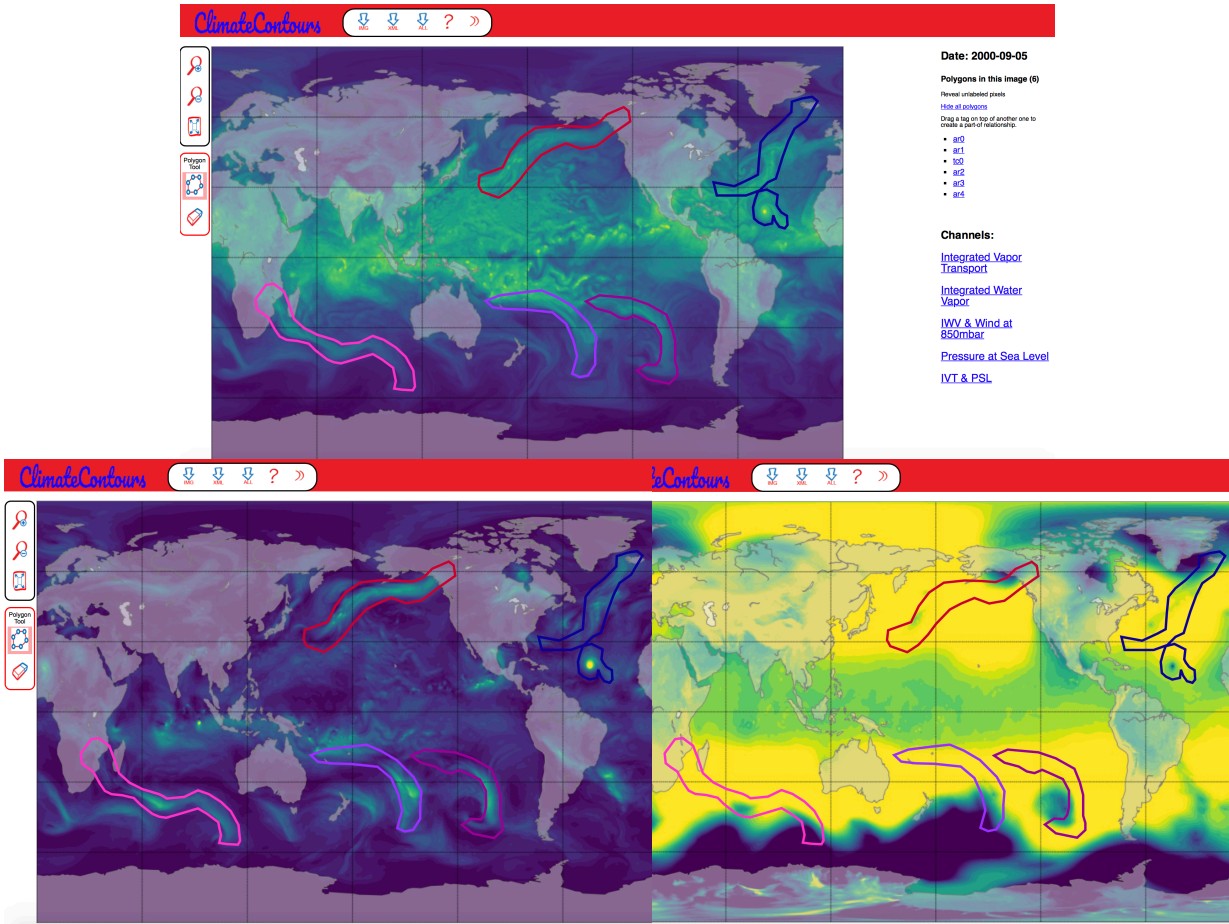

**Figure 1.** The ClimateContours web-based labeling interface. Labelers can choose different channels (physical variables) on the right side of the GUI to display different variables or combinations of variables on the global map. On top: Integrated Water Vapor (IWV) is shown with labels of ARs and TCs; bottom left: Integrated vapor transport (IVT); bottom right: pressure at sea level (PSL).

| Variable | Units |
|---|---|
| Integrated Vapor Transport | $kg\,m^{-1}\,s^{-1}$ |
| Integrated Water Vapor | $mm$ |
| Vorticity | $s^{-1}$ |
| Surface Wind Vectors | $m\,s^{-1}$ |
| 850 hPa Wind Vectors | $m\,s^{-1}$ |
| Sea Level Pressure | $hPa$ |





## 2.2 Labeling Campaigns

In order to capture the expertise of climate scientists in characterizing ARs and TCs, and to obtain sufficient data to train deep neural networks, we conducted multiple labeling campaigns across several institutions and events. These included campaigns at LBNL, UC Berkeley, NCAR, Scripps/UCSD, the 2019 ARTMIP Workshop and the 2019 Climate Informatics Workshop.

For each labeling campaign, participants were briefed on how to use the ClimateContours tool and provided some background on the specifics of ARs and TCs and how to label them effectively. Overall, approximately 80 weather and climate scientists participated in the campaigns and contributed several hundred labelled snapshots of climate data. The ClimateNet dataset currently contains over 1000 carefully curated data labeled by experts using the ClimateContours tool (see Section 2.4 for information about the quality control process). The labeling campaigns proved to be invaluable for not only for generating

high-quality labeled data, but also for obtaining feedback on the ClimateContours tool itself, variables of interest, and how the labeling process could be improved.

## 2.3 Diversity of Expert Labels

Just as there exists a dozen different heuristics for detecting weather events such as atmospheric rivers, tropical cyclones and extra-tropical cyclones (Shields et al., 2018a; Walsh et al., 2010; Neu et al., 2013), we find differences in the labels provided

by experts using the ClimateContours tool. This is perhaps not unexpected as different experts inherently conceptualize and identify weather events in slightly different ways. The labeling campaigns shed useful light on the diversity of labeling styles and implicit assumptions of different experts, as is seen in Figure 2. Disagreements and disparities were most common on the exact spatial extents of individual storms, and less so on the presence or absence of the storm. Some experts disagreed on edge cases, such as incipient events or those that were dissipating. However, we note that the disagreements between human

labels was less severe than differences noticed in dedicated heuristic-based event detection intercomparison projects such as ARTMIP, TCMIP and IMILAST, which exhibit significant disparities in the presence or absence of labeled extreme events and their boundaries (Shields et al., 2018a; Wehner et al., 2018; Ullrich and Zarzycki, 2017; Neu et al., 2013; Walsh et al., 2010).

In Figure 2, labels from 15 different experts are shown. Most experts agree on some of the prominent ARs and TCs, albeit, with some variance in the precise boundaries. The two ARs in the southern Atlantic Ocean and the TC off the west coast of

India are examples of strong expert agreement. However, there are also quite a few discrepancies. A few labelers considered there to be ARs off the east coast of Australia while most did not consider these patterns to represent ARs. Some of the smaller cyclonic structures in the equatorial Pacific also demonstrate discrepancies. One egregious error in labeling can be seen from the triangular AR polygon sitting on the equator in the Atlantic, which was removed in the QA/QC curation process.

## 2.4 Quality Assurance and Quality Control

Any manual labeling campaign, even one conducted amongst experts, is subject to errors stemming from various sources: human errors (lack of expertise/understanding, lack of motivation/thoroughness, mis-interpretation of instructions, mislabeled events, missed events, fatigue) and technical errors (glitches in the web interface, infrastructure). It is simply unrealistic for us



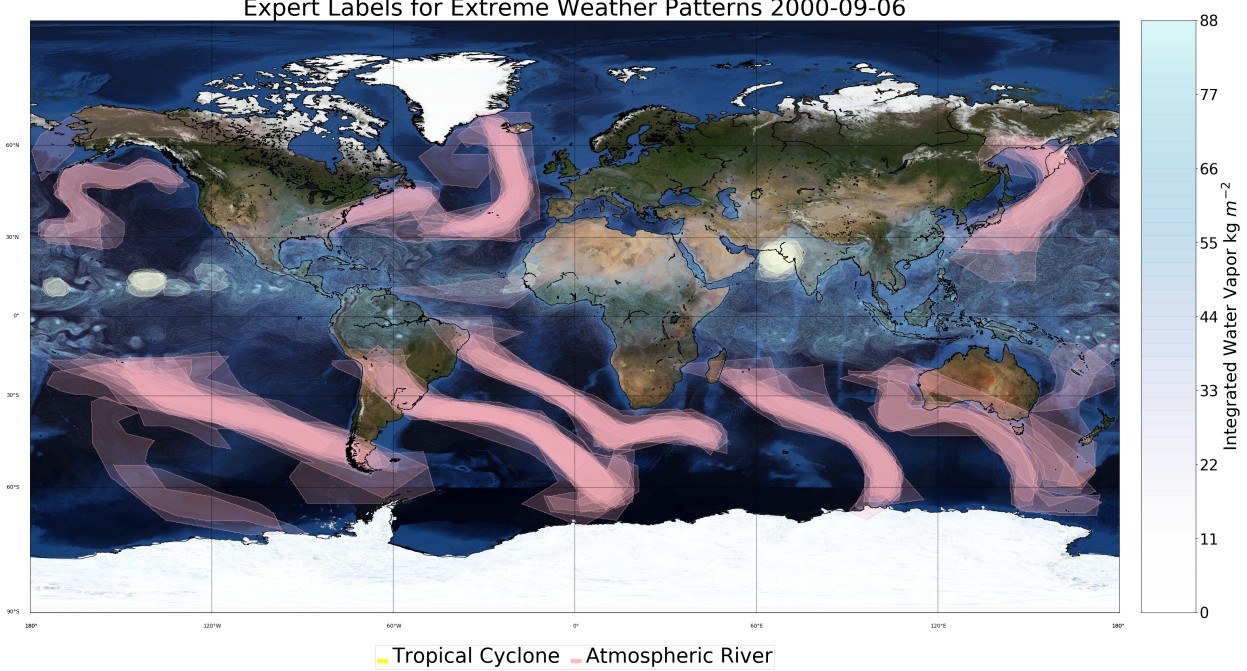

**Figure 2.** Comparison of 15 different expert labelings. Density of pink masks show overlap of AR labels, density of yellow masks show overlap of TC labels. The "bluemarble" map in the background included via Matplotlib's Basemap library is ©NASA.

to expect that all images will be labeled to a consistently high degree of accuracy. In order to address this important issue, we formed a small team of QA/QC experts from the co-author list on this paper. The experts had a background in both climate and computer science; had a good working knowledge of TC and AR patterns, and were briefed on, and motivated to reach a high target accuracy for the labeled dataset. This core team manually examined and executed a thorough "Quality Assurance (QA)

5 / Quality Control (QC)" processes on about 500 samples to correct for errors.

The top priority for the QA/QC team was to fix mis-labeled and missed events. A second type of QA/QC task was to modify the boundaries of correctly labeled events based on an internal consensus grounded in the basic defining characteristics such as: (i) TCs exist in the tropics and are sufficiently intense, measured by low sea level pressure and high vorticity; (ii) ARs sometimes are associated with extra-tropical cyclones (ETCs) but the ETCs should not be included in the AR boundary; (iii)

10 slight differences exist in AR signatures in IWV and IVT fields, and we choose boundaries based on geometric criteria, *i.e. ARs are long, narrow, and transient corridors of strong horizontal water vapor transport* ...(AMS - AR Definition).

Despite making such QA/QC adjustments to experts' labels, there remained some variety amongst AR and TC labels, perhaps representative of the lack of a clear theoretical and quantitative definition for these events. We argue that these relatively minor differences are not a detriment to the training and evaluation of the DL model, as will be shown in the results section next.





# 3   Methods

## 3.1   Deep Learning for Segmentation

In this section, we present our deep learning approach to generate high-quality segmentation masks (i.e. separating objects of interest from the background) for ARs and TCs, using the curated ClimateNet dataset. We model this problem as a semantic

5   segmentation task, i.e. the goal is to assign a class label to every pixel for a given input image. In our case, the input image is the CAM5.1 25-km grid comprising of atmospheric fields, and the output class labels are TC, AR and background (BG).

### 3.1.1   Model

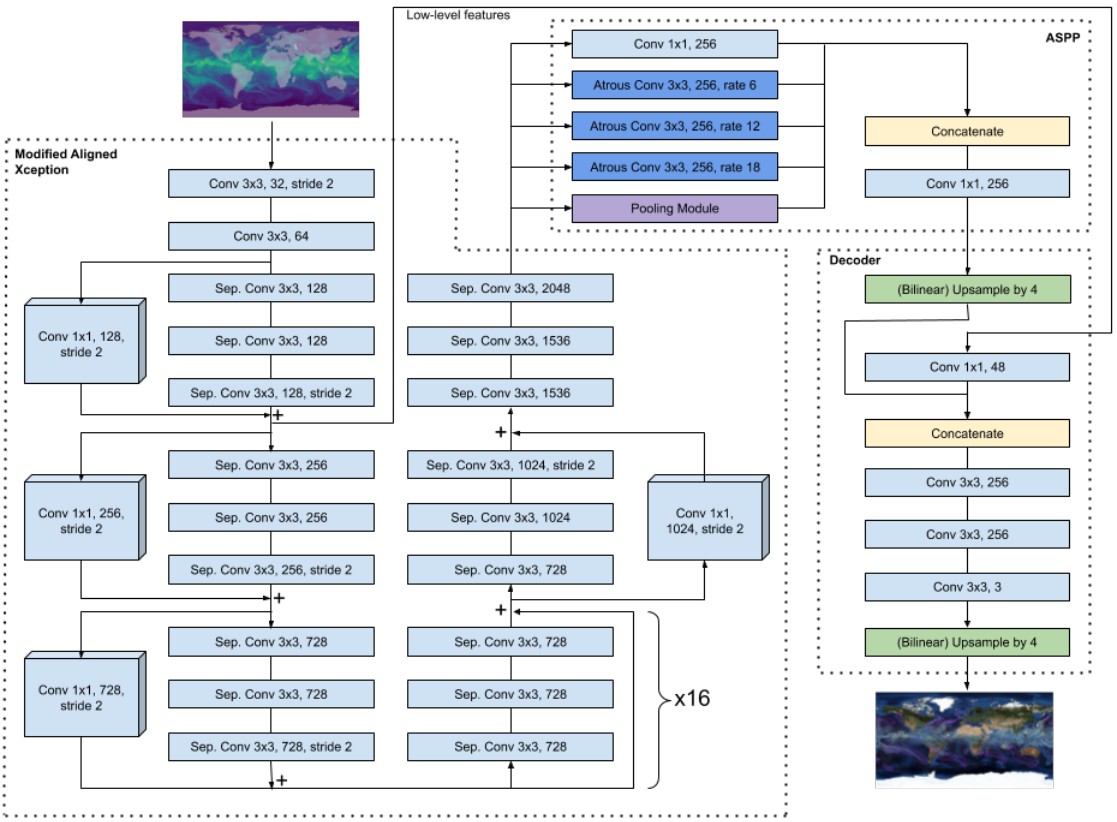

**Figure 3.** DeepLabv3+ network: All convolutional layers are followed by a batch normalization and a ReLU activation layer, which are omitted from this schematic for the sake of brevity. "Sep. Conv" denotes depthwise separable convolution. The pooling module consists of a two-dimensional pooling layer, followed by a convolutional layer, a batch normalization layer and a ReLU activation layer. For more details, we refer the reader to Chen et al. (2018).





We use a PyTorch implementation of DeepLabv3+ (*https://github.com/MLearing/Pytorch-DeepLab-v3-plus*). The input to the model consists of an array of size $(4, 1152, 768)$. It contains atmospheric data from four different channels; namely: TMQ (Total vertically integrated precipitable water), U850 (Zonal wind at 850 mbar pressure surface), V850 (Meridional wind at 850 mbar pressure surface) and PRECT (total convective and large-scale precipitation rate). The deep neural network architecture

used in this work is the DeepLabv3+ architecture (see Figure 3) developed by Chen et al. (2018) based off of Chollet (2016). This architecture has attained state-of-the-art results across various semantic segmentation benchmarks in the computer vision community (PASCAL VOC 2012 and Cityscapes). DeepLabv3+ consists of an *encoder* which captures rich semantic information across multiple scales. The idea of an *encoder* is a series of learnt, hierarchical filters that extracts useful information as pertaining to the task defined - in this case, segmenting ARs and TCs from background. The *decoder* module then upsamples

or in other words, goes from a lower to higher resolution to produce refined object boundaries. There are learnable weights associated with both the *encoder* and the *decoder* which are learnt together while trying to minimize the loss. The loss for the network we use is essentially the $L_2$ difference between the predicted masks and the ground truth labels for a given input.

The output, as discussed earlier, is a segmentation mask of size $(1152, 768)$, where each element in the mask takes the value of either $0$ (BG), $1$ (TC) or $2$ (AR).

### 3.1.2  Training

We study the learning capabilities of DeepLabv3+ on datasets $D_1$ and $D_2$, which correspond to two CAM5.1 scenarios - (i) *All-hist* and (ii) the so-called *UNHAPPI* (Wehner et al. (2018); Mitchell et al. (2017); Wehner et al. (2014)). The *All-hist* scenario runs from 1995-2015 and includes all natural and anthropogenic forcings. The Half a degree Additional warming, Prognosis and Projected Impacts (HAPPI) experimental protocol was designed to compare the effects of stabilizing anthropogenic

global warming at 1.5°C and 2.0°C over preindustrial levels Mitchell et al. (2017). The *UNHAPPI* scenario a stabilizes the anthropogenic warming at 3 °C over preindustrial levels. The details of both scenarios can be found in the listed papers.

We sampled 219 unique images from $D_1$, and used them for labeling campaigns. Each image is labeled by *at least* one human expert, that is, there also exist samples which are labeled by multiple experts. A total of 459 images were acquired upon the completion of the labeling campaigns. The training set for dataset $D_1$ contains 422 (92%) samples; validation and test sets

contain 18 (4%) and 19 (4%) samples, respectively.

A unique challenge in applying standard computer vision-based DL architectures to climate problems is that climate images are heavily imbalanced: 94% of the pixels in ClimateNet data correspond to the "background" class. The DL architecture can naively learn a mapping of any input image pixel to the background class, and be correct 94% of the time! In order to account for this unique challenge, we train our network by optimizing the weighted cross-entropy loss function using the Adam optimizer

(Kingma and Ba, 2014). For such a loss, the class weights are usually defined to be the inverses of class frequencies. However, this choice for the class weights leads to certain numerical issues (Kurth et al., 2018). In order to circumvent these issues, we use the squared inverses of class frequencies as class weights.

We use a learning rate scheduler that multiplicatively reduces the learning rate each time the performance on the validation set does not improve for 3 epochs in a row, and set the initial learning rate to be $1.5 \times 10^{-3}$. We distribute the training process





over 8 GPUs, and use a batch size of 16. We intialize the model with random weights and stop the training as soon as the model's performance on the validation set starts degrading. This corresponds to a training time of 5 epochs for dataset $D_1$.

During training, we track the loss incurred by the model on the training and validation sets. We note that while the model did begin to incur larger losses on the validation set, the incurred training loss never converged. From this observation, we
conclude that the model has potential to learn the segmentation masks even better, if provided with more hand-labeled data.

## 3.2 Inference

Once the training phase is over, we obtain model $M_1$ which was trained on $D_1$; and run inference on held-out samples from the same dataset. We also use the model $M_1$ to run inference on a completely different dataset $D_2$, which corresponds to a climate change scenario. We used a single GPU for this process as inference is computationally much more lightweight than training.
As described above, the models produce segmentation masks of size $(1152, 768)$ for every sample in their respective test sets. These masks are then used to evaluate the performance of the model. A detailed discussion of the results is reported in Section 4.2.

## 3.3 Conditional precipitation analyses

Once Deep Learning has been applied to obtain pixel-level segmentation masks for TCs and ARs, a host of downstream
analytics can now be conducted, for example, we can extract and summarize various *conditional* probability distributions associated with individual event types. In this paper we report on global precipitation associated with TCs and ARs and regional precipitation associated with ARs in the state of California, and TCs in the Gulf of Mexico. Further, we present percentiles and scaling relationships due to global warming of extreme precipitation associated with TCs and ARs at global and regional scales.
A key challenge in conducting such highly precise analytics is the requirement to create conditional probability distributions over $O(10M)$-$O(10)B$ pixels, where each pixel contains the value of a physical quantity at a grid point in the climate model output. We leverage the fastKDE package, developed by O'Brien et al. (2014, 2016b), to compute these distributions efficiently and effectively in seconds to minutes on a single workstation.





## 4 Results

### 4.1 ClimateNet Dataset

The curated expert-labeled ClimateNet dataset, the trained DL segmentation model and PyTorch code to use the model in inference mode are available for download at *https://portal.nersc.gov/project/ClimateNet/*.

### 4.2 Segmentation Results

#### 4.2.1 Qualitative Assessment

We compare visually the performance of the DL model $M_1$ trained on human expert labels in Figure 4. These images illustrate that: (i) DL models are effective at learning mappings between input images and output pixel masks that exist in the training data, *i.e.* they faithfully emulate the data they are trained on; and (ii) the DL model $M_1$ predicts high quality segmentation masks for TCs and ARs that are temporally consistent, even though the notion of temporal persistence of TCs and ARs is not incorporated into the training process. We encourage readers to examine rendered movies at: *https://tinyurl.com/unhappi-yt* which showcase the realism and temporal stability of the segmented TCs and ARs. While there are a few false positives and false negative events, all strong TCs and ARs are successfully detected, segmented and tracked by the DL model $M_1$.

#### 4.2.2 Quantitative Assessment

We measure the performance of our model using the mean Intersection-over-Union (IoU) metric. Given two binary segmentation masks the IoU is defined as the ratio of 'the area of the intersection of two segmentation masks' to 'the area of their union', as illustrated in Figure 5. While it is a measure of the agreement between two masks, it can be far from unity, especially for masks that are small in size because small disagreements in their overlap are amplified by this measure. Hence we emphasize that IoU not be confused with accuracy. Nevertheless, it is a useful metric for evaluating how well a DL model emulates the characteristics of the data it is trained on. Since we have multiple classes in our study, we calculate a mean IoU metric as the mean of the IoUs of the three classes, *i.e.* AR, TC and background.

| IoU Comparisons | | | | |
|---|---|---|---|---|
| | mean IoU | Background IoU | TC IoU | AR IoU |
| Model trained on ClimateNet, $M_1$ | 0.5247 | 0.9389 | 0.2441 | 0.3910 |
| Mean IoU between human experts | 0.5120 | 0.9382 | 0.2567 | 0.3412 |

**Table 1.** After training on ClimateNet, model $M_1$ performs similar to human experts. The overall mean IoU is limited by the TC IoU.

Table 1 shows comparisons of IoUs obtained for the DL model (DeepLabv3+) trained on labels from human experts (first row) and between human experts (second row). In the first row the IoU is calculated on a 'held out' test set that has not been seen by the model during training or validation. For model $M_1$, the IoU is calculated between the model predictions and every

**Figure 4.** Comparison of expert labels from the ClimateNet dataset (left column) and DL segmentation model predictions (right column). The "bluemarble" map in the background included via Matplotlib's Basemap library is ©NASA.





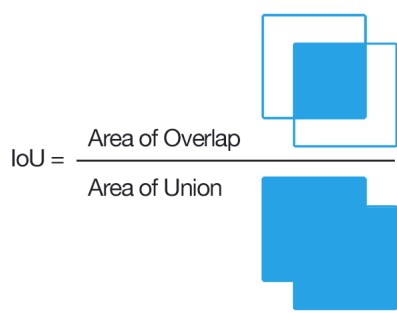

**Figure 5.** Schematic that shows IoU of two square masks. IoU is defined as the ratio of 'the area of the intersection of two segmentation masks' to 'the area of their union'. Note that for the two squares shown here, even an 80% overlap of their edges results in an IoU of 0.47, because the intersection area is 0.64 units and the union is 1.36 units. Source: *https://www.pyimagesearch.com/2016/11/07/intersection-over-union-iou-for-object-detection/*.

human expert label that exists for that image (note that the number of human expert labels are not the same for each image), and then averaged. In the second row we calculate, pair-wise, the IoU between every pair of human expert labels for a given image, and then average over all images. The second row gives a measure of how well any two experts agree on their labels, and we use this as the target metric that the DL model $M_1$ aims to achieve, *i.e.* we train the model to perform similarly to a

human expert.

Given clear, deterministic ground truth labels, where the exact boundaries of every event of each class are well-defined and not subject to discrepancies or uncertainties, the mean IoU is a useful quantitative metric for assessing the quality of segmentation techniques. In our context, however, because the boundaries of ARs and TCs can be hard to define exactly with certainty, it is useful to compare human experts against each other to obtain a measure of the mean IoU between any two human

experts, before evaluating performance of the Deep Learning model against human experts.

In Figure 6 we show an example of the comparison between two human experts for one snapshot. The background class is most dominant because TCs and ARs occupy a small fraction of the total number of pixels on any given image, hence IoUs for background tend to be quite high, as seen in Table 1. However, for TCs and ARs, the IoU can drop to significantly lower values because minor differences in event boundaries for small events can result in low IoU values, as illustrated in Figure 5.

Even though the experts appear to agree reasonably on their event labels and masks, the mean IoU for these two human experts is 0.59. These results are comparable to those reported in Kurth et al. (2018).

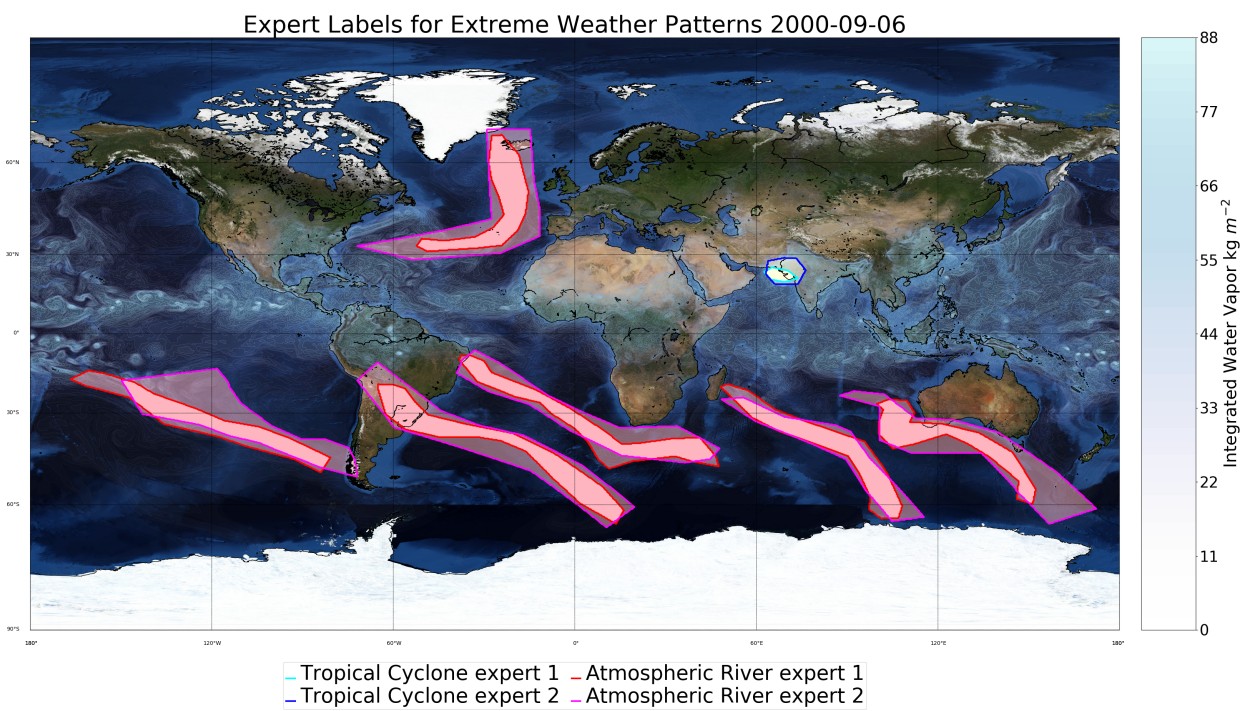

**Figure 6.** Comparison of two different expert labelings with an IoU of 0.59. Note that even though, visually, the experts appear to agree to a large extent on their labels of AR and TC events in this snapshot, the quantitative IoU metric is only 0.59. Hence we emphasize that IoU not be confused with accuracy; good agreement even amongst expert labelers can result in IoU values far from unity. A perfect match, *i.e.* IoU = 1 only results when two labelers agree on every single pixel of the image. As is apparent from this figure, even relatively minor differences in the labels for TCs can disproportionately impact the mean IoU. The "bluemarble" map in the background included via Matplotlib's Basemap library is ©NASA.





### 4.3 Conditional Precipitation Results

One of the main implications of pixel-wise segmentation for climate science is the ability to conduct highly precise analyses conditional on event types, for example, one could as the question, *"How might extreme precipitation due to land-falling atmospheric rivers change in California due to climate change?"*. Here we show some examples of such analyses using
precipitation data.

#### 4.3.1 Global Tropical Cyclone Precipitation

First we calculate annual precipitation from tropical cyclones across the globe for both climate scenarios, All-Hist and the so-called UNHAPPI scenario, by extracting precipitation at every pixel within TC segmentation masks from all datapoints (50 years of data from All-Hist and UNHAPPI). We note that these are average daily rainrates for a 25-km model at 3-hourly
timesteps. Figure 7 shows how tropical cyclone precipitation intensifies and increases in a warmer world. In line with previous studies (Wehner et al., 2018), we see that the PDF of TC precipitation shifts to higher rainrates under global warming. In Table 2 we show the percentiles and rainrates for extreme precipitation from TCs (annual and global). We compare the actual scaling of extreme precipitation with the Clausius-Clapeyron (CC) scaling rate of 7% per K, the scaling rate of available precipitable water in highly saturated atmospheres (O'Gorman and Schneider, 2009; Pall et al., 2007; Allen and Ingram, 2002). Notably,
these studies found that global mean precipitation increases tend to be lower than the increases in the extremes due to different controlling physical mechanisms for each. The tropical (40S-40N) mean SST increase between All-Hist and UNHAPPI, is 1.6K. If extreme tropical cyclone precipitation were to follow a CC scaling relationship, we would expect about an 11.2% increase in extreme precipitation in the warmer simulation. In the last column of Table 2 we see that extremes at the 95th percentile and above scale at super-CC rates. These findings are consistent with hurricane extreme event attribution studies
(Risser and Wehner, 2017; Patricola and Wehner, 2018; van Oldenborgh et al., 2017; Wang et al., 2018) and other idealized tropical analyses (O'Gorman and Schneider, 2009).

| Scaling of TC precipitation under climate change | | | |
|---|---|---|---|
| percentile | precipitation (All-Hist) [mm/day] | precipitation (UNHAPPI) [mm/day] | percentage increase |
| 90 | 46 | 51 | 11.1 |
| 95 | 100 | 116 | 15.5 |
| 99 | 379 | 442 | 16.5 |
| 99.9 | 1010 | 1163 | 15.1 |
| 99.99 | 1476 | 1683 | 14.0 |

**Table 2.** Scaling relationships for global Tropical Cyclone precipitation at various percentiles for extremes. The tropical (40S-40N) mean SST increases by 1.6K from 297.4K (All-Hist) to 299.0K (UNHAPPI). CC scaling for this temperature increase would be 11.2%. Note that extreme precipitation at the 95th percentile and higher exceeds CC scaling with increases about 15%.



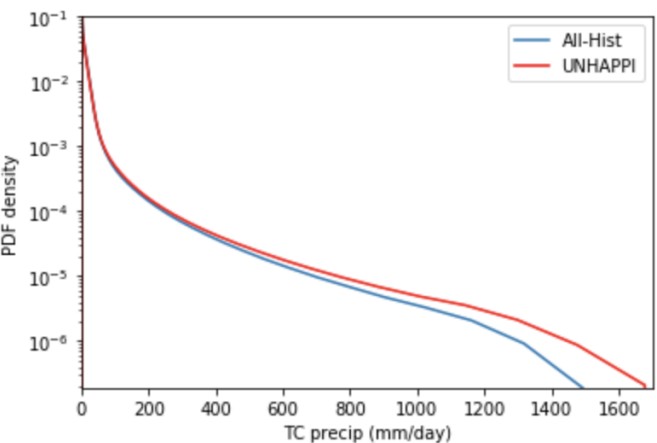

**Figure 7.** Conditional PDF for Tropical Cyclone precipitation computed using fastKDE (O'Brien et al., 2016a)

### 4.3.2 Global Atmospheric River Precipitation

Here we present annual precipitation from atmospheric rivers across the globe for both climate scenarios, All-Hist and UN-HAPPI. Figure 8 shows how AR precipitation intensifies and increases in a warmer world. In line with previous studies (Warner et al., 2015; Espinoza et al., 2018; Gershunov et al., 2019; Gao et al., 2015), we see that the PDF of AR precipitation shifts to
5    higher rainrates.

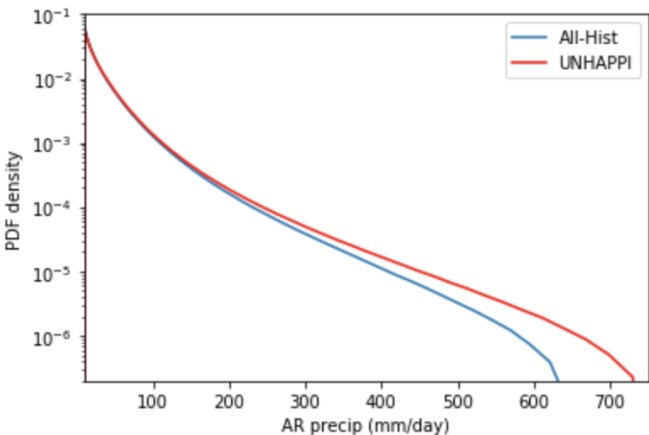

**Figure 8.** (a) Conditional PDF for Atmospheric River precipitation computed using fastKDE (O'Brien et al., 2016a)

In Table 3 we show the percentiles and rainrates for extreme precipitation from ARs (annual and global). The mean SST for AR zones (mid-latitudes, *i.e.* 30S-60S, 30N-60N) increases from 284.9K (All-Hist) to 286.6K (UNHAPPI). Hence, for this 1.7K increase in the reference temperature for ARs, CC scaling implies a 11.9% in precipitation. The actual percentage





| Scaling of AR precipitation under climate change | | | |
|---|---|---|---|
| percentile | precipitation (All-Hist) [mm/day] | precipitation (UNHAPPI) [mm/day] | percentage increase |
| 90 | 42 | 44 | 3.6 |
| 95 | 67 | 70 | 4.3 |
| 99 | 148 | 159 | 7.6 |
| 99.9 | 336 | 378 | 12.7 |
| 99.99 | 603 | 688 | 14.1 |

**Table 3.** Scaling relationships for Atmospheric River precipitation at various percentiles for extremes. Note that extreme ARs have more extreme precip in a warmer world. The mean SST for regions where ARs are most dominant (mid-latitudes, *i.e.* 30S-60S, 30N-60N) increases by 1.7K from 284.9K (All-Hist) to 286.6K (UNHAPPI). Hence CC scaling implies a 11.9% in precipitation.

increases are shown in the last column, and we see that AR precipitation increases scale less than CC for UNHAPPI vs. All-Hist below the 99th percentile, but more than CC for the most extreme events (at and above the 99th percentile). Gao et al. (2015) found projected increases in precipitation from ARs are primarily due to thermodynamic effects controlled by CC, while dynamical effects work counter to this increase for North America. Furthermore, as can be seen in the monotonic

increase in scaling percentages across percentiles, precipitation in stronger ARs intensifies more than weaker ARs in a warmer world (compared to All-Hist). For comparison, Warner et al. (2015) saw mean winter precipitation increase by 11%-18% for the west coast of North America under RCP8.5 while for extreme IVT days, which are closely associated with ARs in this region, precipitation increases by 15%-39%. The findings here are consistent with Warner et al. (2015) although our reported percentages are lower, potentially due to Warner et al. (2015) examining winter time precipitation in California, which shows

robust projected increases (Swain et al., 2018), whereas we examine across the entire year globally.

We now address two questions that highlight the power of pixel-wise segmentation in making localized, precise statements about tropical cyclones and atmospheric rivers in the USA.

### 4.3.3 Tropical Cyclone Precipitation in Gulf of Mexico

First, we focus on the Gulf of Mexico and examine how TC precipitation changes in this region due to global warming. Once again, we calculate the PDFs of precipitation and changes in percentiles of extreme precipitation. The percentage increase in extreme precipitation corresponding to different percentiles are shown in Table 4. The increase in the Gulf of Mexico's temperature between All-Hist (299.5K) and UNHAPPI (301.3K) is 1.8 K, which corresponds to a CC scaling of 12.6%. The last column of this table shows that extreme precipitation due to TCs in the Gulf of Mexico scales well above CC, up to almost

3 times the CC scaling for the most extreme events. Table 4 also suggests that extreme Atlantic hurricanes that are formed in or enter the Gulf of Mexico rain much more intensely compared to global TC trends (illustrated in Figure 7). Further, we examine the number of TC days (defined as a day when at least one TC is active within the specified region, here, the Gulf





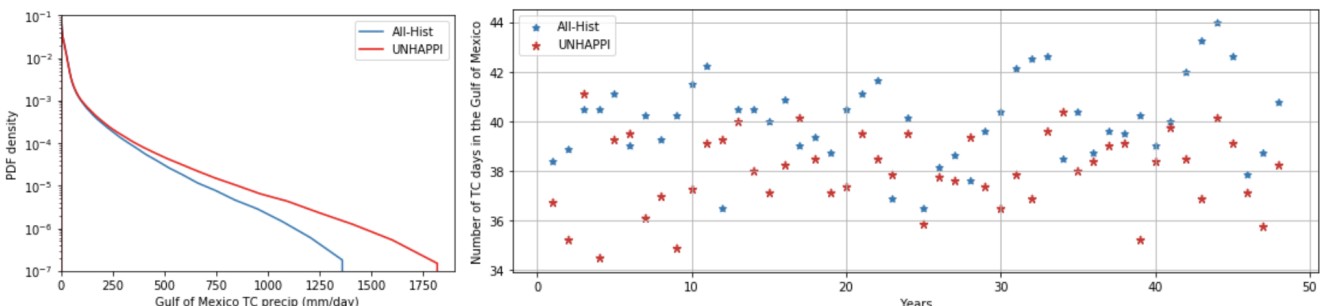

**Figure 9.** (a) Conditional PDF for Tropical Cyclone Precipitation in the Gulf of Mexico using fastKDE (O'Brien et al., 2016a); (b) Number of Tropical Cyclone days annually in the Gulf of Mexico, shown for 50 years of All-Hist and UNHAPPI.

| Scaling of TC precipitation in the Gulf of Mexico under climate change | |
|---|---|
| percentile | percentage increase (UNHAPPI vs. All-Hist) |
| 90 | 19.7 |
| 95 | 21.7 |
| 99 | 31.4 |
| 99.9 | 35.7 |
| 99.99 | 37.4 |

**Table 4.** Scaling relationships for extreme TC precipitation at various percentiles in the Gulf of Mexico. The increase in mean SSTs in the Gulf of Mexico between All-Hist (299.5K) and UNHAPPI (301.3K) is 1.8 K, which results in a CC scaling of 12.6%. We see that extreme precipitation here scales well above CC scaling, corroborated by other studies (Risser and Wehner, 2017; van Oldenborgh et al., 2017; Wang et al., 2018).

of Mexico) in both climate scenarios. In line with what is expected for Atlantic hurricanes under global warming (Wehner et al., 2018; Knutson et al., 2019), we find that, on average, the number of TC days per year decreases from 40.1 (All-Hist) to 37.7 (UNHAPPI). Hence, total precipitation increases by 18.8% per TC day, suggesting that the fewer TCs in the warmer UNHAPPI simulations produce much more precipitation than the cooler All-Hist.

5 ### 4.3.4 Atmospheric River Precipitation in California

Next we focus on California and examine how AR precipitation changes in this region due to global warming. We choose California as ARs play a critical role in California; they can deliver 50% of the annual precipitation but also be a threat to public safety and infrastructure through extreme events (Dettinger et al., 2011).

Once again, we calculate the PDFs of precipitation and changes in percentiles of extreme precipitation. The percentage
10 increase in extreme precipitation corresponding to different percentiles are shown in Table 5. The increase in SST off the coast of California between All-Hist (288.4K) and UNHAPPI (289.7K) is 1.3 K, corresponding to CC scaling of 9.1%. Note that for





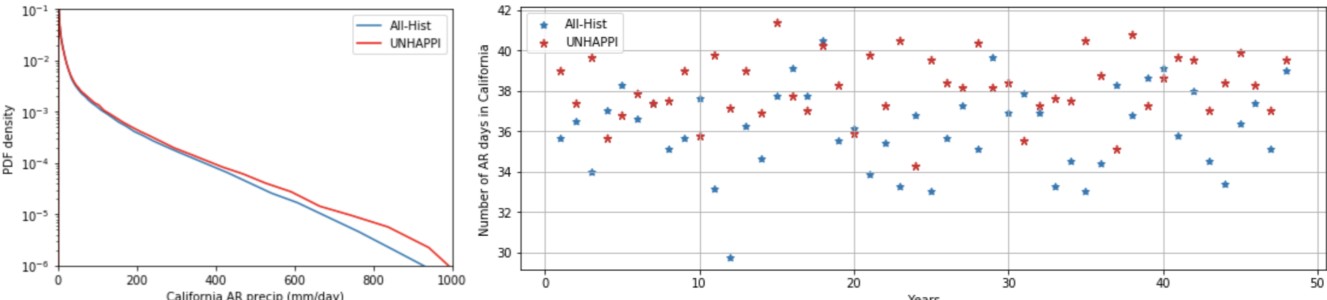

**Figure 10.** (a) Conditional PDF of Atmospheric River Precipitation in California using fastKDE (O'Brien et al., 2016a); (b) Number of Atmospheric River days annually in California, shown for 50 years of All-Hist and UNHAPPI.

| Scaling of AR precipitation in California under climate change | |
|---|---|
| percentile | percentage increase (UNHAPPI vs. All-Hist) |
| 90 | 12.8 |
| 95 | 11.4 |
| 99 | 11.7 |
| 99.9 | 13.2 |
| 99.99 | 10.3 |

**Table 5.** Scaling relationships for extreme AR precipitation at various percentiles in California. The increase in SST off the coast of California between All-Hist and UNHAPPI is 1.3 K, which corresponds to a CC scaling of 9.1%. Note that extreme precipitation scales at super-CC.

all percentiles presented in Table 5, we observe super-CC scaling of extreme precipitation from California ARs. These findings are similar to those of Gao et al. (2015) and Ralph et al. (2017).

We also examine the number of AR days (defined as a day when at least one AR is active within the specified region, here, California) in both climate scenarios. We find that, on average, the number of AR days per year increases from 36.1 (All-Hist)

5 to 37.9 (UNHAPPI). However, total precipitation increases by 36.9% per AR day, suggesting that west coast ARs tend to produce much more precipitation in UNHAPPI. These findings are consistent with a global analysis of ARs under climate change by Espinoza et al. (2018) and regional analysis by Swain et al. (2018). Swain et al. (2018) found projected increases in California's extreme precipitation event frequency. Furthermore, they found these increases to occur during the core winter months and decrease outside of these months. Espinoza et al. (2018) found fewer individual AR events under climate change

10 but an increase in AR conditions globally. Both Espinoza et al. (2018) and Massoud et al. (2019) report AR conditions to increase by 50% and AR IVT strength to increase by 25%. This growth in AR conditions is linked to the increase in both size and IVT intensity of individual AR events. The change in AR days found here falls below the 50% reported by Espinoza et al. (2018) and Massoud et al. (2019) as their AR condition frequency calculations were done at a grid level which is relatively



more sensitive to the increased length and width of ARs compared to our metric. Regardless of AR strength or size, if it makes

contact with California, it will register as an AR day.



## 5 Conclusions and Future Work

We have demonstrated conclusively that Deep Learning models trained on curated expert-labeled climate data – ClimateNet – are powerful tools for segmenting extreme weather patterns in climate datasets, enabling precision climate data analytics. We have developed an end-to-end infrastructure for acquiring expert-labeled data (via ClimateContours); curating the data carefully (using rigorous QA+QC protocols); training DL segmentation models; running DL segmentation models in inference mode; and conducting downstream conditional precipitation analyses.

The proposed dataset – ClimateNet – and end-to-end infrastructure provides several unique capabilities: (i) it enables us to perform fine-grained highly precise data analytics, such as examining changes in frequency and intensity of weather patterns at specific geographic locations across the globe; (ii) it can be applied to different climate scenarios and different datasets without tuning since it does not rely on threshold conditions unlike heuristic algorithms currently used in the community; (iii) the method is suitable for rapidly analyzing large amounts of climate model output. Further, the method can likely be used directly with reanalyses products or observational data using transfer learning, as shown successfully for a similar DL-based method by Ham et al. (2019). While we do not explicitly test the transferability of this model to observations and reanalyses products, we intend to pursue this in future work.

Our work highlights the advantages of transitioning to modern, data-driven DL methods for hi-precision climate data analytics. While our preliminary results are promising, we highlight current limitations in our methodology and identify opportunities for future studies:

1. *Limited Training Data:* The quality of our segmentation results is fundamentally limited by access to large amounts of expert-labeled data. We have only been able to curate ≈500 expert-labelled images thus far, and while the resulting DL model performs reasonably on held-out datasets, we expect that the performance will be improved further with larger amounts of curated expert-labelled data. We appeal to the climate science community to contribute labels to the ClimateNet project – an open source, community project – which is live and freely usable by anyone world-wide at *https://www.nersc.gov/research-and-development/data-analytics/big-data-center/climatenet/*.

   – *Applicability of Transfer Learning:* We intended to leverage the relatively large amount of training data available via AR and TC pattern detection heuristics, such as TECA (Prabhat et al., 2015a), and a smaller amount of ClimateNet labeled data via Transfer Learning (Zamir et al., 2018) to train a model first on heuristics-based training samples followed by "fine-tuning" using ClimateNet data. This approach, however, produced a model that was less skillful at segmenting ARs and TCs compared to a model trained purely on ClimateNet data and further work is needed to understand whether alternative transfer learning techniques are required to obtain more accurate results.

   – *Applicability of Curriculum Learning*: It has been shown that curriculum learning (Lotter et al. (2017); Weinshall et al. (2018)), a type of learning process where a DL model learns to perform well on simpler tasks first before progressively learning harder tasks, is an effective approach for learning complex tasks with limited data. We intend





to employ such techniques, for example, by training on cropped centered snapshots of single events before learning on fully global hi-resolution datasets, and design curricula for efficient and effective learning with limited data.

– *Applicability of Active Learning to prioritize images for labeling:* Our current procedure for choosing candidate climate datapoints for experts to label is unweighted and at random. In particular, we do not choose 'easy' vs. 'hard' images, nor does labeling $N$ images inform the choice of the $(N+1)^{th}$ image presented to a human expert for labeling. In the future, we intend to explore adaptive strategies for downselecting and prioritizing images for manually intensive labeling.

2. *Spatio-Temporal Segmentation:* Our current segmentation models are purely spatial in nature, and do not take temporal persistence of weather events into account. It is, indeed, quite remarkable how well these purely spatial models perform in tracking coherent features through time. To minimize false positives and false negatives and capture more faithfully the evolution of these coherent structures, we intend to augment DL models with consecutive snapshots. There are, however, implications for acquiring expert labels for many more datapoints, and accommodating and training large DL models on GPUs that may require data parallelism and/or model parallelism.

3. *Assessing performance on other types of Climate Data:* Thus far, we have only trained and tested our DL models on CAM5.1, 25-km resolution data. We intend to systematically explore whether the trained model can be applied to: (i) CAM5 output at different spatial and temporal resolutions; (ii) other weather and climate models at comparable resolutions; and (iii) observational and reanalyses products. Given that Deep Learning models learn complex feature representations at multiple levels of abstraction, they will likely work well across modalities, but this generalization claim needs to be tested explicitly.

4. *Probabilistic Segmentation:* We currently acquire labels for every climate snapshot from many human experts with self-ratings on their level of confidence (high, medium, or low) for every event (TC or AR). We do not, however, incorporate these self-ratings into the training procedure, for example, as a form of uncertainty. Building on the work of Mahesh et al. (2019), we intend to use multiple expert labels weighted suitably by their self-ratings for every event to predict pixel-wise probabilistic segmentation masks.

5. *Hypothesis Testing:* Thus far, we have presented preliminary results on changes in extreme precipitation, and associated CC-scaling relationships. One of the unique capabilities provided by our framework is the possibility of rapidly exploring hypotheses related to dynamical mechanisms. For instance, we can index into dynamical variables such as moisture convergence on a per-storm basis; correlate that information with precipitation, temperature and winds; and test hypotheses regarding local dynamical mechanisms being responsible for super-CC scaling. More advanced versions of hypothesis testing could relate dynamical interactions between jet-streams, extra-tropical cyclones and atmospheric rivers.



*Code and data availability.* The data and source code are available at NERSC Science Gateway: https://portal.nersc.gov/project/ClimateNet/.

*Competing interests.* The authors declare that they have no conflict of interest.

*Acknowledgements.* This research used resources of the National Energy Research Scientific Computing Center, a DOE Office of Science User Facility supported by the Office of Science of the U.S. Department of Energy under Contract No. DE-AC02-05CH11231. The
research was performed at the Lawrence Berkeley National Laboratory for the U.S. Department of Energy under Contract No. DE340AC02-05CH11231. Partial support from the Regional and Global Climate Modeling program of the Office of Science, Office of Biological and Environmental Research of the U.S. Department of Energy is gratefully acknowledged. The National Center for Atmospheric Research is sponsored by the National Science Foundation under Cooperative Agreement No. 1852977. This document was prepared as an account of work sponsored by the United States Government. While this document is believed to contain correct information, neither the United States
Government nor any agency thereof, nor the Regents of the University of California, nor any of their employees, makes any warranty, express or implied, or assumes any legal responsibility for the accuracy, completeness, or usefulness of any information, apparatus, product, or process disclosed, or represents that its use would not infringe privately owned rights. Reference herein to any specific commercial product, process, or service by its trade name, trademark, manufacturer, or otherwise, does not necessarily constitute or imply its endorsement, recommendation, or favoring by the United States Government or any agency thereof, or the Regents of the University of California. The views
and opinions of authors expressed herein do not necessarily state or reflect those of the United States Government or any agency thereof or the Regents of the University of California. We would like to acknowledge the labeling campaign organizers and all of the experts who took the time to contribute their expertise towards preparing the ClimateNet dataset.



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
