# Peer review of "ClimateNet: an expert-labelled open dataset and Deep Learning architecture for enabling high-precision analyses of extreme weather"

_Geoscientific Model Development, 2020_

## Referee Comment (RC1) · Anonymous Referee #1 · 24 Apr 2020

Review of GMD-2020-72 ClimateNet: an expert-labelled open dataset and Deep Learning architecture for enabling high-precision analyses of extreme weather

**Summary**

This article introduces ClimateNet an open source, hand-labeled dataset of segmentation maps for tropical cyclones (TC) and atmospheric rivers (AR). It also introduces the interactive ClimateContours tool which enables experts to label TCs and ARs using a web-based interface, together with several data labelling campaigns used to encourage experts to provide labels. The data from ClimateNet was used to train a DeepLabV3+ image segmentation model, which was then applied to CAM 5.1 simulations produce

detailed predictions of severe weather statistics.

**Strengths**

Overall, my impression of this work is very positive, and I feel the article should be accepted for publication in GMD. While tropical cyclones are reasonably amenable to simple heuristics, atmospheric rivers and extratropical cyclones are much harder to quantify. An implicit definition comprised of expert labelled examples avoids the fragility and arbitrariness of hand-crafted heuristics. While the presence or absence of AR and TCs is fairly simple to detect, the pixel-level segmentation masks provided by this effort are quite difficult to come by.

The advantage of pixel-level segmentation of extreme events is made abundantly clear in section 4.3 where conditional precipitation events under a half-degree additional warming scenario are quantified. By integrating precipitation over the predicted segmentations, the authors were able to make detailed predictions for statistical trends in AR and TC intensity and frequency broken down by region and storm severity.

The article is well written. To my eye, the figures, references, tables, and quality metrics seem clear.

**Weaknesses**

While the effort is commendable, the 500 expert labelled images collected thus far is not large, and models trained on this limited dataset are bound to have limited accuracy. The intermediate quality of the predictions is clear from prevalence of false positive TC contours visible in the videos at https://tinyurl.com/unhappi-yt. It is important to obtain a much larger curated set, or to use other techniques to augment it, before detailed predictions can be used with confidence.

Furthermore, the limited dataset size reflects the fact that hand-labelled expert data is extremely scarce. While this effort represents a good start, it seems more work is needed to better leverage this resource. For example, instead of labelling many images

from scratch, the current set could be used to make predictions, and expert time could then be expended to correct those predictions, allowing them to label far more images with less effort. Such human-in-the loop training has been applied in other areas and is one way to better make use of an expert's time.

———————————————————

---

## Referee Comment (RC2) · Imme Ebert-Uphoff (Referee) · 15 May 2020

This manuscript describes a Herculean effort to develop and test a labeled data set for tropical cyclones and atmospheric rivers. The data set consists of multi-channel images, obtained from climate model outputs, along with boundaries (in form of segmentation masks) for TCs and ARs. The manuscript describes not only how the data set was generated, but also demonstrates its utility for climate science analytics. Topics discussed include 1) Setting up a user interface that allows atmospheric scientists to input boundaries, 2) Getting atmospheric scientists to participate and quality control process, 4) Using the resulting data set to train a neural network, and confirming that

the NN trained on this data set actually performs better than those trained on heuristic labels, 5) Applying the NN to climate model outputs of future projections to automatically identify TCs and ARs, and then analyzing their statistics to show amount of projected increase of events, temporal extend and corresponding precipitation worldwide and for specific areas.

Overall, this manuscript describes a huge amount of work, is solid, and provides to the community a dataset that I believe will accelerate research progress regarding research in tropical cyclones, atmospheric rivers, and other atmospheric phenomena. I applaud the team for investing so many resources into creating this important data set, and seeing this effort through. While the number of labels is still relatively small, it's already very useful, and hopefully this article will motivate more atmospheric scientists to contribute a few hours to this effort.

===============================================================
Comments and Questions:

The article has many references, but would benefit from a more thorough analysis of existing work on labeling / detecting ARs and tropical cyclones, be that using heuristics or DL. Please expand that section. Here are some references that come to mind:

1) Bonfanti, C., Trailovic, L., Stewart, J., & Govett, M. (2018, July). Machine Learning: Defining Worldwide Cyclone Labels for Training. 2018 21st International Conference on Information Fusion (FUSION) (pp. 753-760). IEEE. https://ieeexplore.ieee.org/document/8455276

2) C Bonfanti, J Stewart, S Maksimovic, D Hall, M Govett, L Trailovic, I Jankov Detecting Extratropical and Tropical Cyclone Regions of Interest (ROI) in Satellite Data using Deep Learning AGU abstract Dec 2018 https://ui.adsabs.harvard.edu/abs/2018AGUFM.H31H1992B/abstract

**1 is a good demonstration of how labels are difficult to obtain and #2 is complimentary**

to your methods of region detection.

P. 5, Line 14. You say "The placement of vertices ceases when a convex hull is created, i.e. when the last vertex coincides with the first vertex." Do you really mean to say "convex hull", or maybe "closed polygon"? Shapes, especially for bounding ARs, are usually not convex (see also Fig. 1).

Fig. 2: The caption speaks of "yellow masks" for TC labels. In my print-out they look white.

Section 3.1.1: I know the model in Section 3.1.1. is neither new, nor the emphasis of this paper. Nevertheless, for the average reader it would be nice to have one more paragraph that explains the functionality of its different elements a bit more intuitively.

Section 3.1.2: You really just use 5 epochs? I guess with so few training samples...

Fig. 4: It's hard to see the labeling and compare it across the let and right column. Could you use a different color theme?

Fig. 6: How about choosing colors that are more different between Expert 1 and Expert 2?

Section 4.3: Great section that nicely demonstrates the benefits of - and potential way of utilizing - the new data set, and corresponding DL model. I would have liked to see in the tables also the overall increase in precipitation, etc., to see how much that differs from increase in precipitation due to ARs/TCs. But that's not crucial.

Section 5: I really like this section. It has lots of excellent thoughts on limitations and different methods to apply, from active learning (may I suggest Claire Moneleoni as a potential collaborator on that topic?) to transfer learning.

I have one comment for the paragraph on Spatio-Temporal Segmentation. I agree that the temporal persistence of weather events could be an excellent criterion you could utilize. However, rather than acquiring expert labels for more datapoints, as you

propose in that paragraph, couldn't you just make this a constraint for your DL method? The simplest solution - Generate labels for several consecutive time steps using your DL method, then compare them, and only report labels that are fairly consistent across time steps? There are many ways to incorporate such constraints. Would be happy to send REFs (e.g., Vipin Kumar's group at U Minn has done a lot of work in that area, e.g., to detect water bodies from satellite images), but I suspect you already have plenty ideas of your own.
* * *

---

## Author Response (AR1)

**Anonymous Referee #1:** This article introduces ClimateNet an open source, hand-labeled dataset of segmentation maps for tropical cyclones (TC) and atmospheric rivers (AR). It also introduces the interactive ClimateContours tool which enables experts to label TCs and ARs using a web-based interface, together with several data labelling campaigns used to encourage experts to provide labels. The data from ClimateNet was used to train a DeepLabV3+ image segmentation model, which was then applied to CAM 5.1 simulations to produce detailed predictions of severe weather statistics.

**Strengths:** Overall, my impression of this work is very positive, and I feel the article should be accepted for publication in GMD. While tropical cyclones are reasonably amenable to simple heuristics, atmospheric rivers and extratropical cyclones are much harder to quantify. An implicit definition comprised of expert labelled examples avoids the fragility and arbitrariness of hand-crafted heuristics. While the presence or absence of AR and TCs is fairly simple to detect, the pixel-level segmentation masks provided by this effort are quite difficult to come by. The advantage of pixel-level segmentation of extreme events is made abundantly clear in section 4.3 where conditional precipitation events under a half-degree additional warming scenario are quantified. By integrating precipitation over the predicted segmentations, the authors were able to make detailed predictions for statistical trends in AR and TC intensity and frequency broken down by region and storm severity. The article is well written. To my eye, the figures, references, tables, and quality metrics seem clear.

**Weaknesses:** While the effort is commendable, the 500 expert labelled images collected thus far is not large, and models trained on this limited dataset are bound to have limited accuracy. The intermediate quality of the predictions is clear from the prevalence of false positive TC contours visible in the videos at https://tinyurl.com/unhappi-yt. It is important to obtain a much larger curated set, or to use other techniques to augment it, before detailed predictions can be used with confidence. Furthermore, the limited dataset size reflects the fact that hand-labelled expert data is extremely scarce. While this effort represents a good start, it seems more work is needed to better leverage this resource. For example, instead of labelling many images from scratch, the current set could be used to make predictions, and expert time could then be expended to correct those predictions, allowing them to label far more images with less effort. Such human-in-the loop training has been applied in other areas and is one way to better make use of an expert's time.

**Our response:** We thank the reviewer for their insightful comments and clear elucidation of the strengths and weaknesses of our submission. We also thank the reviewer for their very positive impression of our work. By highlighting the key motivations and objectives of this work, such as "*An implicit definition comprised of expert labelled examples avoids the fragility and arbitrariness of hand-crafted heuristics*" and "*The advantage of pixel-level segmentation of extreme events is made abundantly clear in section 4.3 where conditional precipitation events under a half-degree*

*additional warming scenario are quantified*" the reviewer has brought to the forefront the importance of this work in their review.

In response to the reviewer's comments regarding the weaknesses of this submission, we fully agree with the reviewer that the current curated dataset size of approximately 500 expert-labeled images is indeed quite small for training a state-of-the-art deep learning model such as DeepLabv3+, which is complex and has many parameters. We also resonate with the reviewer's comment (*Furthermore, the limited dataset size reflects the fact that hand-labelled expert data is extremely scarce*) that obtaining high-quality expert labels is very tedious and time-consuming. Furthermore, the reviewer's recommendation on expanding the size of the ClimateNet dataset and efficient ways of expanding the dataset is very well-received. In fact, that is exactly what we have embarked upon following the submission: instead of having experts label many images from scratch, we are using the trained DeepLabv3+ model's predictions in inference mode (on new images different from the original training data), and then using expert time to correct those predictions. Indeed, this enables experts to correct segmentation masks that are already of high quality, and in particular, delete the false positive TCs and ARs, instead of labeling images from scratch. As the reviewer has rightly pointed out, this "human-in-the loop training" is an effective way to better make use of an expert's time. We anticipate that our dataset size will increase to a few thousand high-quality labeled images in a short period of time. We are confident that with a larger training dataset that will be achieved from the "human-in-the loop training" we will be able to significantly reduce false positives and eradicate the problem of fragmented events.
**Referee #2:** This manuscript describes a Herculean effort to develop and test a labeled data set for tropical cyclones and atmospheric rivers. The data set consists of multi-channel images, obtained from climate model outputs, along with boundaries (in the form of segmentation masks) for TCs and ARs. The manuscript describes not only how the data set was generated, but also demonstrates its utility for climate science analytics. Topics discussed include 1) Setting up a user interface that allows atmospheric scientists to input boundaries, 2) Getting atmospheric scientists to participate, 3) the quality control process, 4) Using the resulting data set to train a neural network, and confirming that the NN trained on this data set actually performs better than those trained on heuristic labels, 5) Applying the NN to climate model outputs of future projections to automatically identify TCs and ARs, and then analyzing their statistics to show amount of projected increase of events, temporal extend and corresponding precipitation worldwide and for specific areas.

Overall, this manuscript describes a huge amount of work, is solid, and provides to the community a dataset that I believe will accelerate research progress regarding research in tropical cyclones, atmospheric rivers, and other atmospheric phenomena. I applaud the team for investing so many resources into creating this important data set, and seeing this effort through. While the number of labels is still relatively small, it's already very useful, and hopefully this article will motivate more atmospheric scientists to contribute a few hours to this effort.

======================================================================

**Our response:** We thank the reviewer for their kind acknowledgement of the efforts and achievements described in our manuscript. We also thank the reviewer for their very positive impression of our work. Furthermore, we are grateful that the reviewer explicitly urges the atmospheric science community to contribute to the expansion of the dataset. Finally, by highlighting some key aspects of this work, such as "*the NN trained on this data set actually performs better than those trained on heuristic labels*" and "*provides to the community a dataset that I believe will accelerate research progress regarding research in tropical cyclones, atmospheric rivers, and other atmospheric phenomena*" the reviewer has brought to the forefront the importance of this work in their review.

======================================================================

**Comments and Questions:**

The article has many references, but would benefit from a more thorough analysis of existing work on labeling / detecting ARs and tropical cyclones, be that using heuristics or DL. Please expand that section. Here are some references that come to mind:

1) Bonfanti, C., Trailovic, L., Stewart, J., & Govett, M. (2018, July). Ma- chine Learning: Defining Worldwide Cyclone Labels for Training. 2018 21st In- ternational Conference on Information Fusion (FUSION) (pp. 753-760). IEEE. https://ieeexplore.ieee.org/document/8455276

2) C Bonfanti, J Stewart, S Maksimovic, D Hall, M Govett, L Trailovic, I Jankov Detecting Extratropical and Tropical Cyclone Regions of Inter- est (ROI) in Satellite Data using Deep Learning AGU abstract Dec 2018 https://ui.adsabs.harvard.edu/abs/2018AGUFM.H31H1992B/abstract

**1 is a good demonstration of how labels are difficult to obtain and #2 is complimentary to your methods of region detection.**

**Our response:** We thank the reviewer for pointing out additional references that we missed out. We will certainly include these references with suitable descriptions in the body of the revised manuscript.
========================================================================

P. 5, Line 14. You say "The placement of vertices ceases when a convex hull is created, i.e. when the last vertex coincides with the first vertex." Do you really mean to say "convex hull", or maybe "closed polygon"? Shapes, especially for bounding ARs, are usually not convex (see also Fig. 1).

**Our response:** We thank the reviewer for catching this nuance -- yes, indeed, we mean closed polygons and agree that these boundaries are not always convex. We will correct this in the revised manuscript.
========================================================================

Fig. 2: The caption speaks of "yellow masks" for TC labels. In my print-out they look white.

**Our response:** We will update this to "light-colored masks" to avoid ambiguities from printed versus online colors.
========================================================================

Section 3.1.1: I know the model in Section 3.1.1. is neither new, nor the emphasis of this paper. Nevertheless, for the average reader it would be nice to have one more paragraph that explains the functionality of its different elements a bit more intuitively.

**Our response:** We thank the reviewer for pointing this out. We will certainly include a para describing the model with some intuition in the revised manuscript.
========================================================================

Section 3.1.2: You really just use 5 epochs? I guess with so few training samples...

**Our response:** Yes, indeed. We suspect that the large image size (768x1152) and the multiple examples of the same class (TC or AR) in each global snapshot provides more information to the NN than one would normally expect from a single image (say in computer vision examples). Perhaps this aids faster convergence of the NN...
========================================================================

Fig. 4: It's hard to see the labeling and compare it across the let and right column. Could you use a different color theme?

Fig. 6: How about choosing colors that are more different between Expert 1 and Expert 2?

**Our response:** We thank the reviewer for pointing out these issues with clarity. We will attempt to fix these in the revised manuscript.
========================================================================

Section 4.3: Great section that nicely demonstrates the benefits of - and potential way of utilizing - the new data set, and corresponding DL model. I would have liked to see in the tables also the overall increase in precipitation, etc., to see how much that differs from increase in precipitation due to ARs/TCs. But that's not crucial.

**Our response:** We thank the reviewer for suggesting additional characteristics of precipitation to examine. This is part of an ongoing detailed investigation of extreme precipitation changes using the DL model for segmentation and full 3-dimensional fields of several other variables (omega, q, T etc.), including attributing changes in extreme precipitation to thermodynamic and dynamic contributions.
========================================================================

Section 5: I really like this section. It has lots of excellent thoughts on limitations and different methods to apply, from active learning (may I suggest Claire Moneleoni as a potential collaborator on that topic?) to transfer learning.

**Our response:** We thank the reviewer for these kind remarks and suggestion. We will follow up with Claire Monteleoni regarding active learning.
========================================================================

I have one comment for the paragraph on Spatio-Temporal Segmentation. I agree that the temporal persistence of weather events could be an excellent criterion you could utilize. However, rather than acquiring expert labels for more datapoints, as you propose in that paragraph, couldn't you just make this a constraint for your DL method? The simplest solution -

Generate labels for several consecutive time steps using your DL method, then compare them, and only report labels that are fairly consistent across time steps? There are many ways to incorporate such constraints. Would be happy to send REFs (e.g., Vipin Kumar's group at U Minn has done a lot of work in that area, e.g., to detect water bodies from satellite images), but I suspect you already have plenty of ideas of your own.

**Our response:** We thank the reviewer for suggesting ways to incorporate temporal persistence of events into the DL model. We would appreciate additional references and suggestions. We are also considering several other approaches and extensions to improve the performance of the DL model, including reducing false positives and true negatives, by incorporating ideas from persistent homology and using 3D space-time convolutions.
========================================================================